# Contribution of GRACE Satellite Mission to the Determination of Orthometric/Normal Heights Corrected for Their Dynamics—A Case Study of Poland

Malgorzata Szelachowska * , Walyeldeen Godah and Jan Krynski

Institute of Geodesy and Cartography (IGiK), Centre of Geodesy and Geodynamics, 27 Modzelewskiego St., 02-679 Warsaw, Poland

* Correspondence: malgorzata.szelachowska@igik.edu.pl; Tel.: +48-22-3291903

**Abstract:** Physical heights were traditionally determined without considering the dynamic processes of the Earth induced from temporal mass variations. The Gravity Recovery and Climate Experiment (GRACE) mission provided valuable data that allow the estimation of geoid/quasigeoid height changes and vertical deformations of the Earth's surface induced from temporal mass loading, and thereby temporal variations of physical heights. The objective of this investigation is to discuss the determination of orthometric/normal heights considering mass transports within the Earth's system. An approach to determine such heights was proposed. First, temporal variations of orthometric/normal heights ($\Delta H/\Delta H^*$) were determined using the release 6 GRACE-based Global Geopotential Models together with load Love numbers obtained from the preliminary reference Earth model. Then, those variations were modelled and predicted using the seasonal decomposition (SD) method. The proposed approach was tested on the territory of Poland. The main results obtained reveal that $\Delta H/\Delta H^*$ over the area investigated are at the level of a couple of centimetres and that they can be modelled and predicted with a millimetre accuracy using the SD method. Orthometric/normal heights corrected for their dynamics can be determined by combining modelled $\Delta H/\Delta H^*$ with orthometric/normal heights referred to a specific reference epoch.

**Keywords:** orthometric/normal heights; GRACE satellite mission; temporal variations of orthometric/ normal heights; temporal variations of geoid/quasigeoid heights; vertical deformations of the Earth's surface

## 1. Introduction

The determination of heights is one of the major tasks in geodesy. Knowledge of heights is needed in various engineering applications as well as in scientific research associated with the Earth's shape and its dynamics. Physical heights, e.g., orthometric/normal heights ($H/H^*$), were traditionally obtained using the expensive and time-consuming precise spirit levelling. Nowadays, they are more and more frequently determined as a combination of ellipsoidal heights from precise satellite navigation techniques with geoid/quasigeoid heights derived from gravity data. Extensive investigations have thus been conducted to determine geoid/quasigeoid heights, ellipsoidal heights, and the resulting orthometric/normal heights of high accuracy. The issue of modelling gravimetric geoid/quasigeoid of a sub-cm accuracy is currently discussed by many authors, e.g., [1–3]. It is also considered as one of the main activities of the International Association of Geodesy [4]. The ellipsoidal heights can be determined with a couple of millimetres accuracy using long GNSS (global navigation satellite system) observation sessions and a high-precision scientific GNSS data processing software [5]. The $H/H^*$ of the mean square error of the adjusted levelling network less than 1 mm/km can be obtained from the first-order levelling network established using spirit levelling.

Excluding the areas of an evident land subsidence/uplift in which the annual change of $H/H^*$ is considered [6–8], $H/H^*$ are, nowadays, practically treated as static heights in the majority of land areas. The modern precise observation techniques indicate, however, that $H/H^*$ are worldwide affected by both annual and semi-annual variations. Heights obtained from repeated levelling measurements are influenced by artifacts and aliasing induced from both secular and seasonal vertical land motion [9]. The seasonality of the ellipsoidal height determined from GNSS data induced by temporal mass loading was demonstrated by numerous authors, e.g., in a global scale [10] and a regional as well as local scale in South America [11], Europe [12], Japan [13], the Amazon basin [14], Bangladesh [15], West Africa [16], North America [17], China [18], Tibet [19], Poland [20–22], and East Africa [23]. The Gravity Recovery and Climate Experiment (GRACE [24]) satellite mission considerably contributed to the improvement of the knowledge concerning temporal mass variations within the Earth's system (TMVES) and thereby the Earth's time-variable gravity field as well as the dynamic processes of the Earth. Based on data from the GRACE satellite mission, changes of the geoid/quasigeoid height in the time domain ($\Delta N/\Delta \zeta$) have been investigated by many authors, e.g., [25–32]. Vertical deformations of the Earth's surface, $\Delta h$, induced from the temporal mass loading have also been investigated using GRACE satellite mission data [12,33]. Godah et al. [30,31], and Öztürk et al. [34] investigated orthometric/normal height changes ($\Delta H/\Delta H^*$) estimated using GRACE data over Central Europe and over Turkey. They reveal that peak to peak variations of $\Delta H/\Delta H^*$ reach 23 mm over Central Europe and 25 mm over Turkey. Godah et al. [35] assessed $\Delta H/\Delta H^*$ induced by hydrological mass transports for twenty-four worldwide large river basins, using GRACE data. They illustrated that for the river basin of a strong hydrological signal (e.g., the Amazon River basin), peak to peak variations $\Delta H/\Delta H^*$ are ca. 80 mm. In general, all these studies reveal the importance of considering $\Delta N/\Delta \zeta$, $\Delta h$, and $\Delta H/\Delta H^*$ in the Earth's science-related disciplines. Moreover, temporal variations of the aforementioned heights remain one of the problems that should carefully be handled for the vertical datum unification [36]. In addition to the GRACE satellite mission, the GRACE-Follow On (cf. https://gracefo.jpl.nasa.gov/ accessed on 4 August 2022) and next generation gravity missions [37] as well as atomic clock measurements [38] are aimed to improve the modelling of TMVES and thereby the monitoring of those temporal heights variations.

Taking into consideration the $\Delta N/\Delta \zeta$ and $\Delta h$ induced from TMVES, correcting $H/H^*$ for their dynamics will essentially be required. The overarching objective of this research is to discuss the determination of orthometric/normal heights corrected for their dynamics considering the effect of TMVES. Four examples illustrating the impact of TMVES on heights are discussed in Section 2. Section 3 describes an approach for the determination of $H/H^*$ corrected for their dynamics. Section 4 illustrates the determination of such heights over Poland as a case study. Finally, conclusions and recommendations are provided in Section 5.

## 2. The Impact of TMVES on Heights

The TMVES interact with height changes in different ways depending mainly on the location of mass changes and whether they exhibit an increase or decrease in mass. Herein, four cases are discussed (see Figure 1). In the first and second cases, the mass change is assumed to be beneath the geoid/quasigeoid surface (Figure 1a,b) [35]. They show that when the mass under the geoid/quasigeoid surface increases/decreases, (1) the gravitational potential also increases/decreases, and thereby the geoid/quasigeoid surface ascends/descends, and (2) the Earth's surface gets deformed upward/downward. In the third and fourth cases illustrated in Figure 1c,d, the mass changes are assumed on the Earth's surface, i.e., above the geoid/quasigeoid surface, such as hydrological mass changes. When the mass on the Earth's surface increases/decreases, the Earth's surface deforms downward/upward and the geoid/quasigeoid surface ascends/descends. In the third and fourth cases, $\Delta H/\Delta H^*$ are inversely proportional to TMVES.

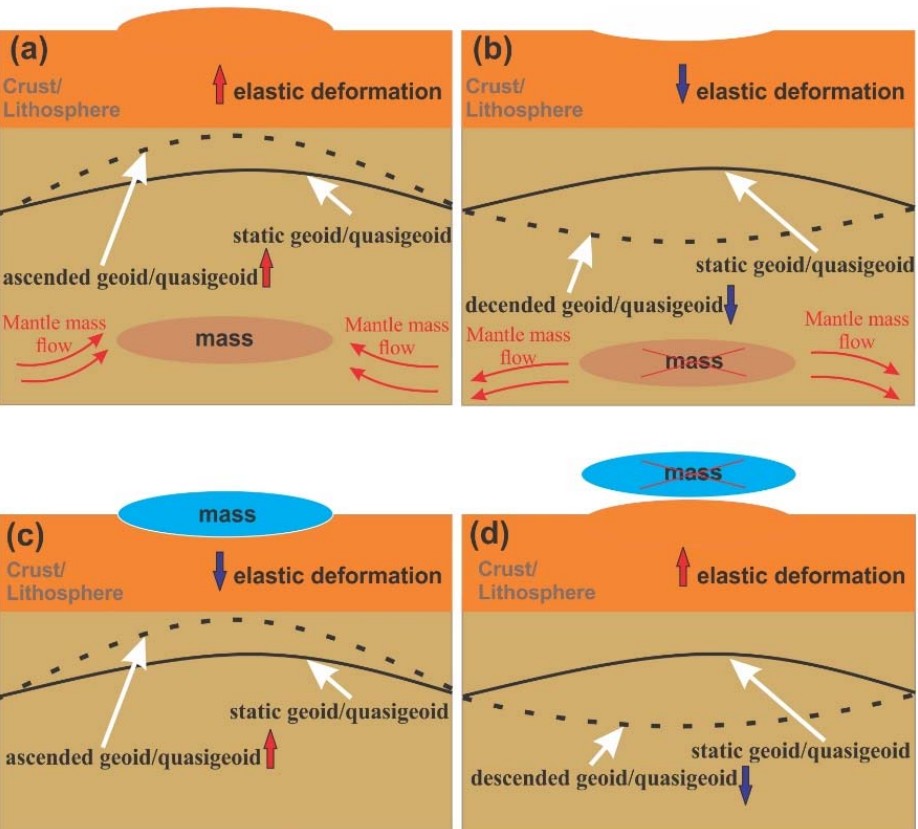

**Figure 1.** The relation between temporal mass variations within the Earth's system (TMVES), geoid/quasigeoid height changes, and the elastic deformation of the Earth's surface in the up component. (**a**) the mass under the geoid/quasigeoid surface increases, (**b**) the mass under the geoid/quasigeoid surface decreases, (**c**) the mass on the Earth's surface increases, and (**d**) the mass on the Earth's surface decreases.

## 3. The Determination of Orthometric/Normal Heights Corrected for Their Dynamics

The orthometric/normal heights corrected for their dynamics $H(t)/H^*(t)$ can be defined at arbitrary time $t$ as a sum of the orthometric/normal height $H(t_0)/H^*(t_0)$ at the reference epoch $t_0$, and temporal variations of orthometric/normal heights $\Delta H(t)/\Delta H^*(t)$ at $t$.

The mathematical formulation of the proposed approach for the determination of $H(t)/H^*(t)$ at epoch $t$ is expressed as follows:

$$H(t) = H(t_0) + \Delta H^M(t) \tag{1}$$

$$H^*(t) = H^*(t_0) + \Delta H^{*M}(t) \tag{2}$$

where $\Delta H^M(t)/\Delta H^{*M}(t)$ are values of $\Delta H/\Delta H^*$ at time $t$ calculated from the $\Delta H/\Delta H^*$ model. This approach is described as follows (cf. Figure 2):

(1) Determination of $H(t_0)/H^*(t_0)$.

(2) Estimation of $\Delta H/\Delta H^*$ over the time interval including $t_0$ and if possible, $t$, by combining $\Delta N/\Delta \zeta$ and $\Delta h$. Then, developing the model of $\Delta H/\Delta H^*$ and calculating $\Delta H^M(t)/\Delta H^{*M}(t)$ from this model.

(3) Determination of $H(t)/H^*(t)$ at $t$ as a sum of $H(t_0)/H^*(t_0)$ and $\Delta H^M(t)/\Delta H^{*M}(t)$ obtained from step 2.

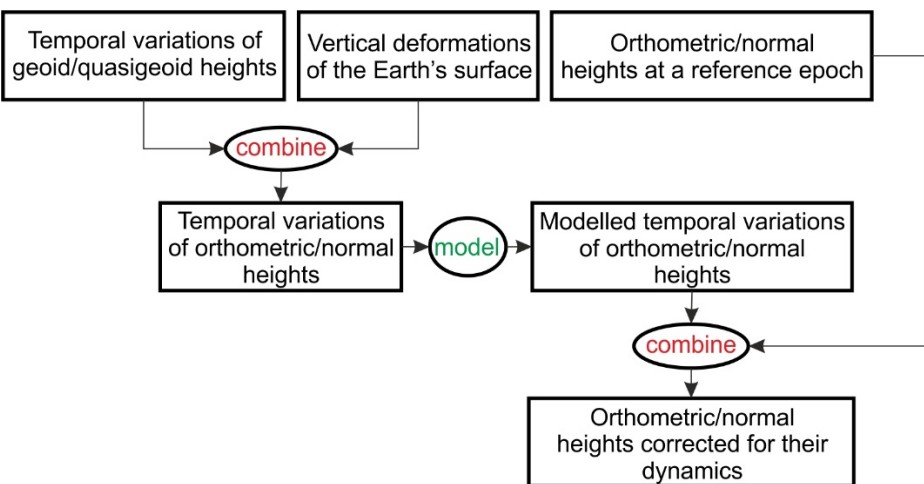

**Figure 2.** Flowchart describing the general steps to determine orthometric/normal heights corrected for their dynamics.

Precise levelling networks, i.e., first order levelling networks, on a national as well as a regional scale, were commonly established by levelling measurement campaigns conducted over a long-time span in different epochs. Thus, a proper reduction of those levelling measurements to a common epoch is applied to determine $H(t_0)/H^*(t_0)$ at the reference epoch $t_0$. Moreover, $H(t_0)/H^*(t_0)$ are, nowadays, determined by combining the ellipsoidal height of high accuracy obtained from GNSS measurements with geoid/quasigeoid height obtained from a precise geoid/quasigeoid model developed with the use of high-quality gravity data. It should be noted that establishing/re-measuring a national precise levelling network usually takes many years. Thus, secular (i.e., linear trend) variations of orthometric/normal heights that occurred during these years must be considered for the establishment of the 1st order levelling network referenced to a specific epoch such as $t_0$. Although $\Delta H(t)/\Delta H^*(t)$ can theoretically be obtained with the use of geometric data, e.g., spirit levelling, in practice, it is not possible to repeat levelling campaigns sufficiently frequently to provide as frequent as monthly solutions for heights [9]. Thus, monthly variations of orthometric/normal heights can practically be estimated using data from dedicated gravity satellite missions, in particular the GRACE/GRACE–FO missions, combined with longer time series data such as repeated levelling, glacial isostatic adjustment (GIA), GNSS time series, and tide gauge records, that also contain variations of physical heights signal. In this study, $\Delta H(t)/\Delta H^*(t)$ are estimated as follows

$$\left.\begin{aligned} \Delta H(t_i^G) &= T^{\Delta H}(t_i^G) + \left(\Delta h(t_i^G) - \Delta N(t_i^G) - T(t_i^G)\right) \quad \text{for orthometric heights} \\ \Delta H^*(t_i^G) &= T^{*\Delta H^*}(t_i^G) + \left(\Delta h(t_i^G) - \Delta \zeta(t_i^G) - T^*(t_i^G)\right) \quad \text{for normal heights} \end{aligned}\right\} \quad (3)$$

where $t_i^G$ ($i = 1, 2, \ldots, n$) is the epoch of the $i$th (of $n$ considered) monthly GRACE/GRACE–FO global geopotential models (GGMs), $T(t_i^G)/T^*(t_i^G)$ present the linear trend of $\Delta H(t)/\Delta H^*(t)$ obtained from GRACE/GRACE–FO data, and $T^{\Delta H}(t_i^G)/T^{*\Delta H^*}(t_i^G)$ denote the linear trend (i.e., secular variation) of $\Delta H(t)/\Delta H^*(t)$, estimated with the use of data from repeated levelling, GIA, GNSS, or tide gauges. It should be noted that when the epoch $t_0$, to which orthometric/normal heights of the levelling network are referenced does not correspond to one of the epochs $t_i^G$, a proper transformation of $H(t_0)/H^*(t_0)$ to the closest $t_i^G$ epoch would be required.

The $\Delta N(t_i^G)$, $\Delta \zeta(t_i^G)$ and $\Delta h(t_i^G)$ can be estimated from GRACE/GRACE–FO–based GGMs as follows [33,39]:

$$\Delta N(t_i^G) = \Delta \zeta(t_i^G) = N\left(t_i^G\right) - N(t_0) = \zeta(t_i^G) - \zeta(t_0) \quad (4)$$

$$\zeta(t_i^G) = \frac{GM}{r\gamma} \sum_{l=2}^{l_{max}} \left(\frac{a}{r}\right)^l \sum_{m=0}^{l} \left(C_{lm}^T\left(t_i^G\right)\cos m\lambda + S_{lm}^T\left(t_i^G\right)\sin m\lambda\right) P_{lm}(\sin\varphi) \tag{5}$$

$$\zeta(t_0) = \frac{GM}{r\gamma} \sum_{l=2}^{l_{max}} \left(\frac{a}{r}\right)^l \sum_{m=0}^{l} \left(C_{lm}^T(t_0)\cos m\lambda + S_{lm}^T(t_0)\sin m\lambda\right) P_{lm}(\sin\varphi) \tag{6}$$

$$\Delta h(t_i^G) = h(t_i^G) - h(t_0) \tag{7}$$

$$h(t_i^G) = \frac{3a\rho_w}{\rho_{avg}} \sum_{l=0}^{l_{max}} \sum_{m=0}^{l} P_{lm}(\sin\varphi) \frac{h_l}{2l+1} \left(C_{lm}^\sigma\left(t_i^G\right)\cos m\lambda + S_{lm}^\sigma\left(t_i^G\right)\sin m\lambda\right) \tag{8}$$

$$h(t_0) = \frac{3a\rho_w}{\rho_{avg}} \sum_{l=0}^{l_{max}} \sum_{m=0}^{l} P_{lm}(\sin\varphi) \frac{h_l}{2l+1} \left(C_{lm}^\sigma(t_0)\cos m\lambda + S_{lm}^\sigma(t_0)\sin m\lambda\right) \tag{9}$$

with

$$C_{lm}^T\left(t_i^G\right) = C_{lm}^W\left(t_i^G\right) - C_{lm}^U, \quad S_{lm}^T\left(t_i^G\right) = S_{lm}^W\left(t_i^G\right) - S_{lm}^U \tag{10}$$

$$C_{lm}^T(t_0) = C_{lm}^W(t_0) - C_{lm}^U, \quad S_{lm}^T(t_0) = S_{lm}^W(t_0) - S_{lm}^U \tag{11}$$

and surface density coefficients $C_{lm}^\sigma(t_i^G)$, $C_{lm}^\sigma(t_0)$, $S_{lm}^\sigma(t_i^G)$ and $S_{lm}^\sigma(t_0)$ defined as

$$\left\{ \begin{array}{c} C_{lm}^\sigma\left(t_i^G\right) \\ S_{lm}^\sigma\left(t_i^G\right) \end{array} \right\} = \frac{\rho_{avg}}{3\rho_w} \frac{2l+1}{1+k_l} \left\{ \begin{array}{c} C_{lm}^T\left(t_i^G\right) \\ S_{lm}^T\left(t_i^G\right) \end{array} \right\}, \tag{12}$$

$$\left\{ \begin{array}{c} C_{lm}^\sigma(t_0) \\ S_{lm}^\sigma(t_0) \end{array} \right\} = \frac{\rho_{avg}}{3\rho_w} \frac{2l+1}{1+k_l} \left\{ \begin{array}{c} C_{lm}^T(t_0) \\ S_{lm}^T(t_0) \end{array} \right\} \tag{13}$$

where $\varphi$, $\lambda$, $r$ are the latitude, longitude, and the geocentric radius of the computation point $Q$ on the physical surface of the Earth, $a$ is the semi-major axis of the reference ellipsoid, $GM$ is the product of the Newtonian gravitational constant and the Earth's mass, $C_{lm}^U$ and $S_{lm}^U$ are spherical harmonic coefficients of the normal gravity field, $C_{lm}^W$ and $S_{lm}^W$ are fully normalized spherical harmonics from GRACE/GRACE–FO–based GGMs, $\gamma$ denotes the normal gravity at $Q$, $P_{lm}$ is a fully normalized Legendre function of degree $l$ and order $m$, $l_{max}$ is the maximum degree applied, $\rho_{avg}$ is the Earth's average density, $\rho_w$ is the water density, and $h_l$ and $k_l$ are the load Love numbers (LLN) of degree $l$.

Geoid heights $N(t_i^G)$ and $N(t_0)$ are obtained using Equations (5) and (6), respectively, considering the reduction of the potential induced by topographic masses, i.e., the topographic bias [40], and assuming $r = R$ and $\gamma = \gamma_0$, where $R$ is the mean radius of the Earth and $\gamma_0$ is the normal gravity at $Q_0$ on the ellipsoid, i.e., the corresponding point of $Q$ projected along the plumb line onto the ellipsoid. The $\Delta N(t_i^G)$ can be considered equal to $\Delta\zeta(t_i^G)$ [35].

The $\Delta H/\Delta H^*$ can be modelled using a suitable method such as the seasonal decomposition (SD) method [41] and the principle component analysis/empirical orthogonal function (PCA/EOF) method [42]. In this research, the SD method, following the algorithm and computation steps described in detail in [29], was applied to model $\Delta H/\Delta H^*$. The $\Delta H^c(t)/\Delta H^{*c}(t)$ time series is obtained by centering the $\Delta H(t)/\Delta H^*(t)$ time series to zero. Then, the $\Delta H^c(t)/\Delta H^{*c}(t)$ is decomposed into a seasonal component $S(t)/S^*(t)$, a long-term component $LT(t)/LT^*(t)$, and an unmodelled residual component $\varepsilon(t)/\varepsilon^*(t)$. The model of temporal variations of orthometric/normal heights $\Delta H^{SD}(t)/\Delta H^{*SD}(t)$ is obtained as a sum of long-term and seasonal components.

$$\Delta H^{*c}(t)/\Delta H^{*c}(t) = LT(t)/LT^*(t) + S(t)/S^*(t) + \varepsilon(t)/\varepsilon^*(t) \tag{14}$$

$$\Delta H^{SD}(t)/\Delta H^{*SD}(t) = LT(t)/LT^*(t) + S(t)/S^*(t) \tag{15}$$

The $\Delta H^M(t)/\Delta H^{*M}(t)$ were determined by referring the $\Delta H^{SD}(t)/\Delta H^{*SD}(t)$ to the reference epoch $t_0$. Then, $\Delta H^M(t)/\Delta H^{*M}(t)$ values were interpolated/extrapolated at the height datum benchmarks investigated. The $H(t)/H^*(t)$ were calculated by combining these interpolated/extrapolated values with $H(t_0)/H^*(t_0)$ at those sites.

In order to mitigate the effect of the gaps in data from the GRACE/GRACE–FO satellite missions and the latency of GRACE/GRACE–FO products, prediction of the $\Delta H^M/\Delta H^{*M}$ values for months where these products are unavailable, would be required. Within this study, the prediction of $\Delta H^M/\Delta H^{*M}$ values were investigated using the empirical approach proposed in Godah et al. [29] based on the results obtained from the SD method. The predicted values of $\Delta H^M/\Delta H^{*M}$ were calculated as a sum of the $S(t)/S^*(t)$ component and the $\Delta H/\Delta H^*$ values obtained from an appropriate mathematical model fitted to $LT(t)/LT^*(t)$ component.

## 4. A Case Study

The area of Poland, as a unique one covered with high-quality terrestrial gravity datasets, GNSS/levelling data, and gravimetric geoid/quasigeoid models, has been chosen as a study area. The accuracy of quasigeoid heights obtained from the recent gravimetric quasigeoid model and GNSS/levelling data as well as combined GGMs, e.g., the EGM2008 (Earth Gravitational Model 2008; [43]), over this area is estimated at the level of 1 cm [44]. Over this area, $\Delta N/\Delta\zeta$ were reliably determined, analyzed, and modelled using GRACE satellite mission data [29], and $\Delta h$ were detected at the ASG-EUPOS (Polish Active Geodetic European Position Determination System; http://www.asgeupos.pl/, accessed on 1 August 2022) network sites using GNSS data [21,22].

### 4.1. Orthometric/Normal Heights at Reference Epoch

In this study, $H(t_0)/H^*(t_0)$ from the ASG-EUPOS network sites were utilized. The values of normal heights referred to the Kronstadt86 vertical datum for the territory of Poland (PL-KRON86-NH) are accessible via the ASG-EUPOS website (http://www.asgeupos.pl/index.php accessed on 1 August 2022). These normal heights were transformed into the PL-EVRF2007-NH (European Vertical Reference Frame 2007 recently applied for the area of Poland). The reference epoch $t_0$ of PL-EVRF2007-NH is 1 January 2008, which corresponds to its realization epoch [45]. The PL-EVRF2007-NH is based on results of the adjustment of the 4th levelling campaign in Poland (1998–2012), EUVN (European Vertical Reference Network), and EUVN-DA (European Vertical Reference Network-Densification Action) solutions as well as solutions for the benchmarks of eccentric points of permanent GNSS stations of the ASG-EUPOS network. Heights are given in the zero tidal system. The accuracy of levelling measurements was estimated as 0.74 mm/km. Based on the adjustment of the levelling network, the estimated accuracy of a single benchmark is 3.5 mm [46].

The orthometric heights at the ASG-EUPOS sites at the reference epoch $t_0$ were determined using the well-known basic relation for the geoid–quasigeoid separation [47]:

$$H^* - H \approx \frac{\Delta g^B}{\overline{\gamma}} H \tag{16}$$

where $\Delta g^B$ is the Bouguer gravity anomaly and $\overline{\gamma}$ is the mean normal gravity along the normal plumb line between the ellipsoid and the telluroid.

The $H$ and $H^*$ of ASG-EUPOS sites referred to the PL-EVRF2007-NH at the reference epoch $t_0$ are shown in Figure 3. It should be noted that the differences between orthometric and normal heights over the area of Poland do not exceed a couple of decimeters [44].

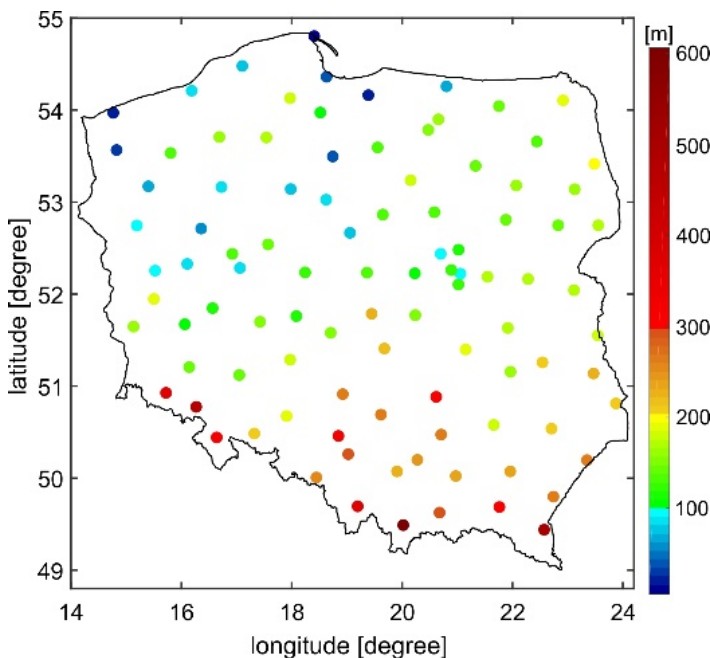

**Figure 3.** Orthometric/normal heights of ASG-EUPOS sites referred to the PL-EVRF2007-NH at the reference epoch 1 January 2008.

### 4.2. Temporal Variations of Orthometric/Normal Heights

Temporal variations of orthometric/normal heights $\Delta H(t)/\Delta H^*(t)$ over Poland induced by TMVES were considered. In Section 4.2.1, temporal variations of an equivalent water thickness $\Delta EWT(t)$ and $\Delta H(t)/\Delta H^*(t)$ over the area investigated were estimated, and the relation between $\Delta EWT(t)$ and $\Delta H(t)/\Delta H^*(t)$ was discussed. Section 4.2.2 demonstrates the development of the $\Delta H(t)/\Delta H^*(t)$ model and computing $\Delta H^M(t)/\Delta H^{*M}(t)$ values from this model. Section 4.2.3 discusses the possibility for predicting $\Delta H^M(t)/\Delta H^{*M}(t)$.

### 4.2.1. The Estimation of $\Delta H/\Delta H^*$

The territory of Poland was divided into equal area blocks of a size equivalent to the spatial resolution of GRACE data, i.e., ~334 × 334 km² or 3° × 3° on the equator. Those equal area blocks are localized in the same configuration as the Jet Propulsion Laboratory (JPL) mascons (mass concentrations), global 3° × 3° equal area spherical cap mascon solutions. Figure 4 shows the area investigated, including the location of four mascons used. The main TMVES over this area can be ascribed to temporal hydrological mass variations [28]. Temporal variations of equivalent water thickness $\Delta EWT(t_i^G)$ over $t_i^G - t_0$ at the location of those mascons obtained from monthly release 06 (RL06) JPL GRACE mascon solutions using the mascon visualization tool (cf. https://ccar.colorado.edu/grace/index.html, accessed on 1 August 2022) are depicted in Figure 5.

The results presented in Figure 5 indicate a distinctive pattern of seasonal water mass variations with minimum and maximum values in July–September and March, respectively. This seasonal variations pattern agrees well with the corresponding one documented in Krynski et al. [28]. It reveals that the decrease/increase in water masses over Poland results from intensive water evaporation during dry months in the summer season and the accumulation of snow/water in the winter season.

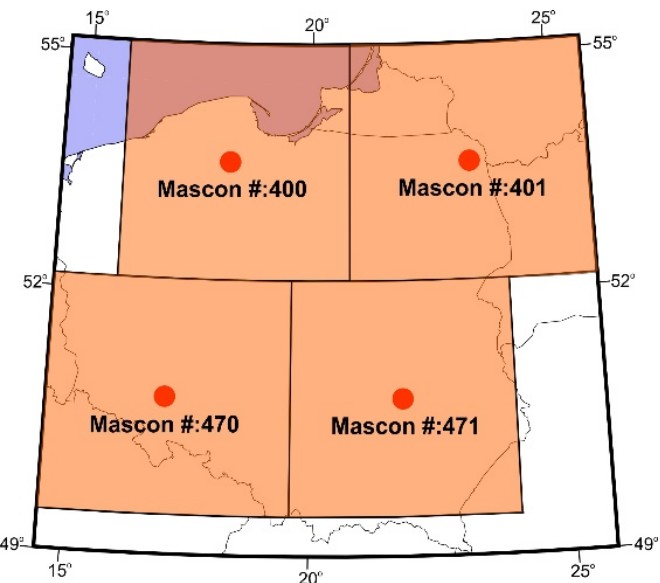

**Figure 4.** The locations of mascons over Poland.

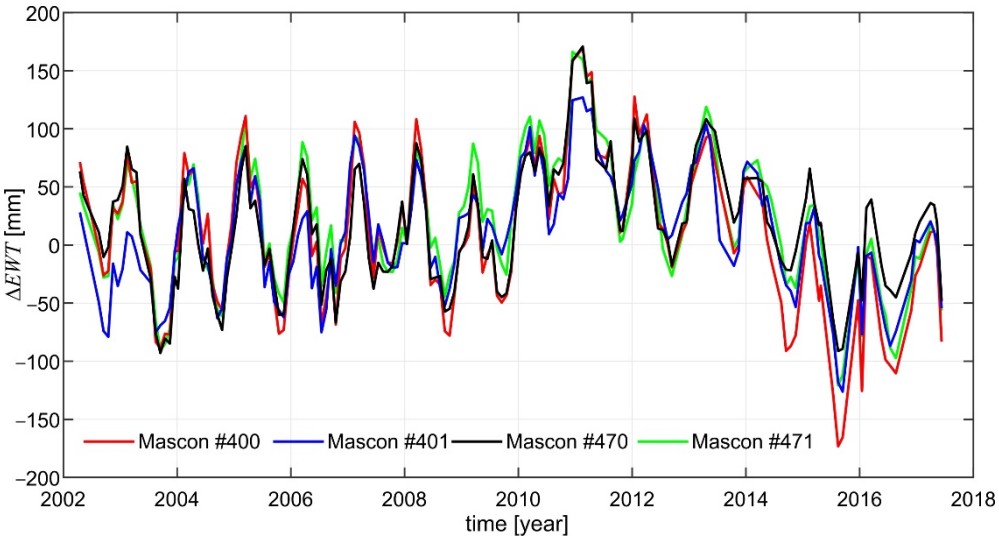

**Figure 5.** Temporal variations of equivalent water thickness ($\Delta EWT$) over Poland.

The $\Delta N(t_i^G)/\Delta\zeta(t_i^G)$, as well as $\Delta h(t_i^G)$, were estimated at the location of mascons (cf. Figure 4) using monthly CSR (Computation Centre of the University of Texas Center of Space Research) RL06 GRACE-based GGMs [48], LLN calculated using the preliminary reference Earth model (PREM; [49]), and the IGiK–TVGMF (Instytut Geodezji i Kartografii–Temporal Variations of Gravity/Mass Functionals; [50]) software. The degree-1 and degree-2 harmonic coefficients from the solution described in Swenson et al. [51] and the corresponding ones obtained from satellite laser ranging (SLR) [52], respectively, replaced those of CSR RL06 GRACE-based GGMs. In order to reduce the noise included in CSR RL06 GRACE-based GGMs, the decorrelation filter DDK3 [53] was utilized. This filter was chosen as it compromises between reducing the noise and keeping the signal over the area investigated [29]. Moreover, those GGMs were truncated at d/o 60 that corresponds to the spatial resolution of the mascon grid (Figure 4) and the DDK3 filter. Then, $\Delta H(t_i^G)/\Delta H^*(t_i^G)$ were estimated as a sum of detrended $\Delta N(t_i^G)/\Delta\zeta(t_i^G)$, detrended $\Delta h(t_i^G)$, and the linear trend (i.e., secular variations) of $H/H^*$ obtained from the NKG2016LU model [54]. The resulting $\Delta N(t_i^G)/\Delta\zeta(t_i^G)$, $\Delta h(t_i^G)$, and $\Delta H(t_i^G)/\Delta H^*(t_i^G)$ for the period April 2002–August 2016 are depicted in Figure 6.

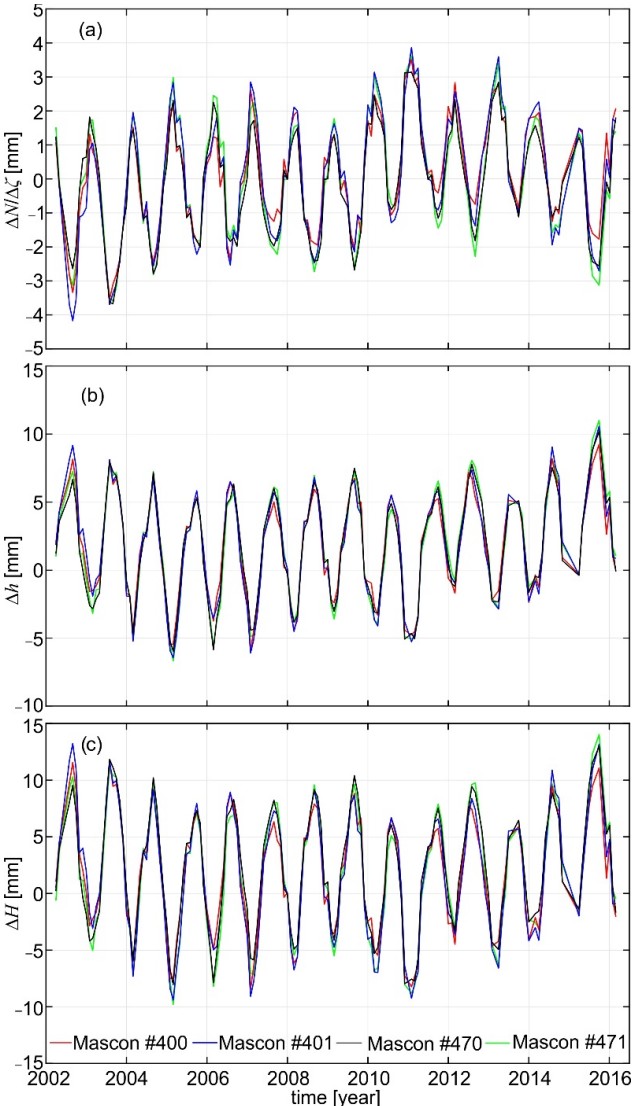

**Figure 6.** Time series of (**a**) geoid/quasigeoid changes ($\Delta N/\Delta\zeta$), (**b**) vertical deformations of the Earth's surface ($\Delta h$), and (**c**) temporal variations of orthometric/normal heights ($\Delta H/\Delta H^*$) over Poland estimated using GRACE-based GGMs.

Figure 6 exhibits a clear seasonal pattern of $\Delta N(t_i^G)/\Delta\zeta(t_i^G)$, $\Delta h(t_i^G)$, and $\Delta H(t_i^G)/\Delta H^*(t_i^G)$. The maximum values of $\Delta N(t_i^G)/\Delta\zeta(t_i^G)$ were obtained at the beginning of spring (in February–April) and minimum values at the end of summer/beginning of autumn (in August–October); they are consistent with the corresponding ones estimated by Krynski et al. [28] and Godah et al. [29,30,32]. For $\Delta h(t_i^G)$, maximum values were obtained in August–October and minimum values in February–April, which correspond to minima and maxima of $\Delta N(t_i^G)/\Delta\zeta(t_i^G)$, respectively. For a subarea, the dispersions of $\Delta N(t_i^G)/\Delta\zeta(t_i^G)$ and $\Delta h(t_i^G)$ reach 8 mm and 17 mm, respectively. Minimum and maximum values of $\Delta H(t_i^G)/\Delta H^*(t_i^G)$, obtained as the combination of $\Delta N(t_i^G)/\Delta\zeta(t_i^G)$ and $\Delta h(t_i^G)$, were observed in February–April and in August–October, respectively. The $\Delta H(t_i^G)/\Delta H^*(t_i^G)$ for the same subarea reach up to 23 mm, and 4 mm from one subarea to another at the same epoch. The results presented in Figure 6 also indicate that approximately 66% of the $\Delta H(t_i^G)/\Delta H^*(t_i^G)$ signal is due to $\Delta h$, while the remaining part of this signal is induced from $\Delta N/\Delta\zeta$. The coefficients of correlation between $\Delta EWT(t_i^G)$ (cf. Figure 5) and $\Delta N(t_i^G)/\Delta\zeta(t_i^G)$, $\Delta h(t_i^G)$, and $\Delta H(t_i^G)/\Delta H^*(t_i^G)$ are presented in Figures 7–9, respectively, showing strong correlations between water mass changes and $\Delta N/\Delta\zeta$ as well as $\Delta h$. The correlations between $\Delta EWT(t_i^G)$ and $\Delta N(t_i^G)/\Delta\zeta(t_i^G)$ are slightly stronger

(correlation coefficients of $0.89 \pm 0.02$) compared to those between $\Delta EWT(t_i^G)$ and $\Delta h(t_i^G)$, characterized by correlation coefficients of $-0.71 \pm 0.04$. This might be due to the fact that $N/\zeta$ is basically associated with the disturbing potential generated from the Earth's mass distribution, while the vertical deformations of the Earth's surface may depend on other factors, e.g., the elasticity of the Earth's crust. Overall, the results obtained reveal that the resulting $\Delta H/\Delta H^*$ are strongly correlated with $\Delta EWT$ (correlation coeffiecients of $-0.80 \pm 0.04$).

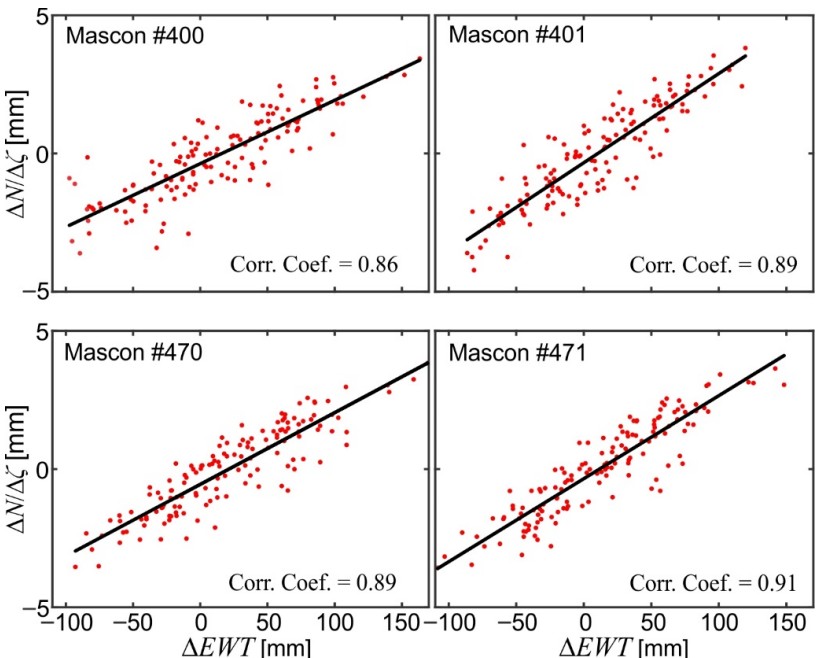

**Figure 7.** Correlations between temporal variations of equivalent water thickness ($\Delta EWT$) and geoid/quasigeoid changes ($\Delta N/\Delta \zeta$) over Poland.

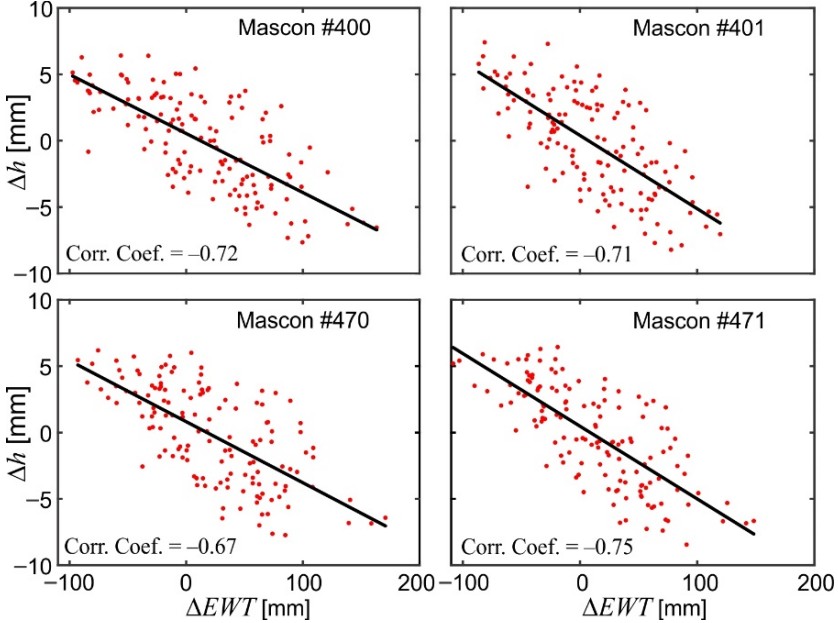

**Figure 8.** Correlations between temporal variations of equivalent water thickness ($\Delta EWT$) and vertical deformations of the Earth's surface ($\Delta h$) over Poland.

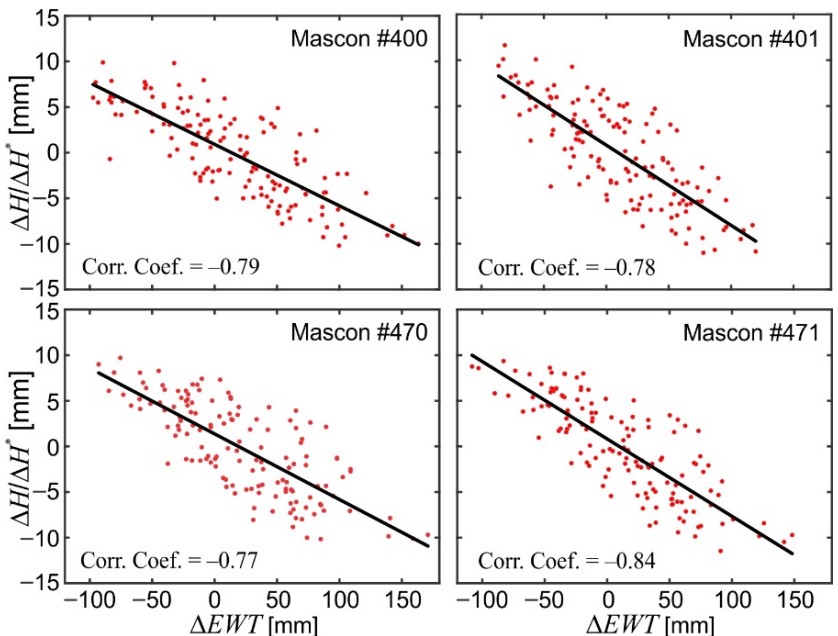

**Figure 9.** Correlations between temporal variations of equivalent water thickness ($\Delta EWT$) and temporal variations of orthometric/normal heights ($\Delta H / \Delta H^*$) over Poland.

### 4.2.2. The Analysis and Modelling of $\Delta H / \Delta H^*$ and Determining $\Delta H^M / \Delta H^{*M}$

The seasonal decomposition (SD) method was used for the analysis and modelling of the $\Delta H / \Delta H^*$ determined. The $\Delta H^c(t_i^G) / \Delta H^{*c}(t_i^G)$ time series, obtained by centering the $\Delta H(t_i^G) / \Delta H^*(t_i^G)$ time series to zero for the period between January 2004 and December 2010 exhibiting no gaps in monthly CSR RL06 GRACE-based GGMs, was applied. The long-term and seasonal components of $\Delta H^{SD}(t_i^G) / \Delta H^{*SD}(t_i^G)$, depicted in Figure 10, were used in Equation (15) for modelling $\Delta H / \Delta H^*$. Figure 11 shows $\Delta H^c(t_i^G) / \Delta H^{*c}(t_i^G)$ and $\Delta H^{SD}(t) / \Delta H^{*SD}(t)$ models developed using the SD method (see Equation (15)). It also illustrates the differences $\delta \Delta H / \delta \Delta H^*$ between $\Delta H^c(t_i^G) / \Delta H^{*c}(t_i^G)$ and their models $\Delta H^{SD}(t) / \Delta H^{*SD}(t)$.

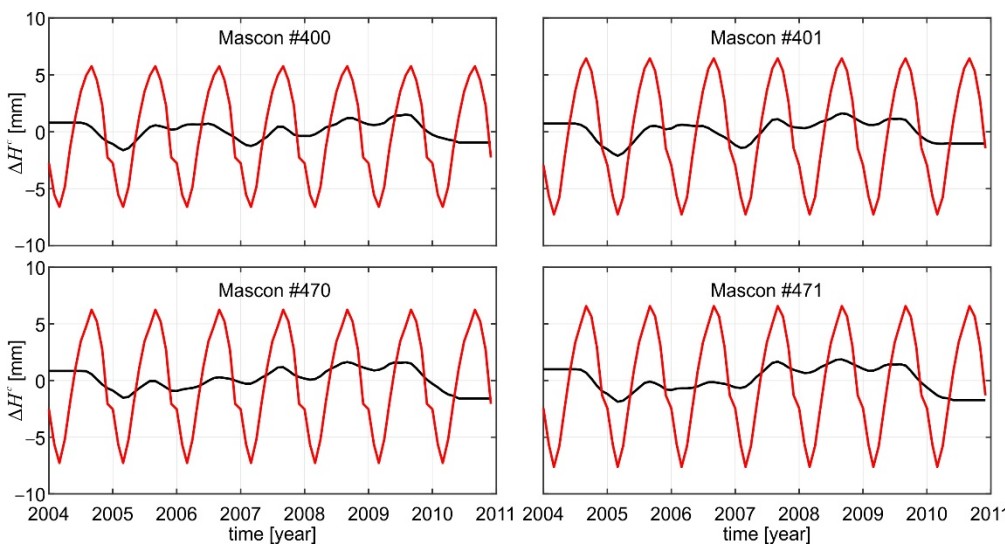

**Figure 10.** Seasonal (red line) and long term (black line) components of $\Delta H^c(t_i^G) / \Delta H^{*c}(t_i^G)$ over Poland.

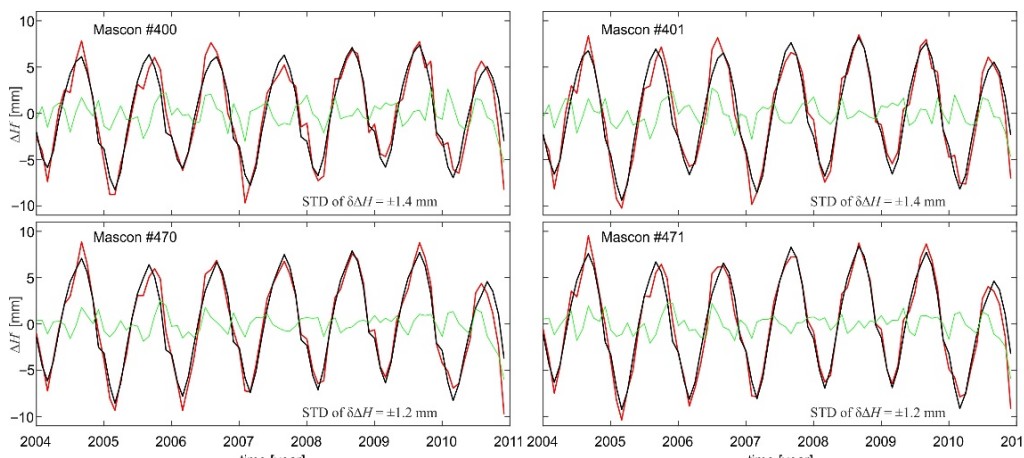

**Figure 11.** Time series of $\Delta H^c(t_i^G)/\Delta H^{*c}(t_i^G)$ (black line) and $\Delta H^{SD}(t)/\Delta H^{*SD}(t)$ models obtained using the SD method (red line), and differences between $\Delta H^c(t_i^G)/\Delta H^{*c}(t_i^G)$ and their corresponding $\Delta H^{SD}(t)/\Delta H^{*SD}(t)$ models (green line).

Figure 10 indicates that the dominant signal of temporal variations of orthometric/normal heights comes from seasonal components of $\Delta H^c(t_i^G)/\Delta H^{*c}(t_i^G)$. The amplitudes of these seasonal components reach the level of $6.5 \pm 0.5$ mm. The long-term components of $\Delta H^c(t_i^G)/\Delta H^{*c}(t_i^G)$ for the area and time period investigated exhibit a non-linear changing pattern. They fluctuate within the range of $\pm 2$ mm. The results presented in Figure 11 indicate a good agreement between $\Delta H^{SD}(t)/\Delta H^{*SD}(t)$ models and $\Delta H^c(t_i^G)/\Delta H^{*c}(t_i^G)$ data. The coefficients of correlation between $\Delta H^{SD}(t)/\Delta H^{*SD}(t)$ and the respective $\Delta H^c(t_i^G)/\Delta H^{*c}(t_i^G)$ are at the level of 0.97. The standard deviations of $\delta \Delta H/\delta \Delta H^*$ are in the range of $1.3 \pm 0.1$ mm.

The $\Delta H^{SD}(t)/\Delta H^{*SD}(t)$ models are centered to zero (see Figure 11). Thus, in order to determine $\Delta H^M(t)/\Delta H^{*M}(t)$, the $\Delta H^{SD}(t)/\Delta H^{*SD}(t)$ had to be referred to the reference epoch $t_0$, i.e., 1 January 2008, that has been used as the realization epoch of orthometric/normal heights. In this study, $\Delta H^M(t)/\Delta H^{*M}(t)$ were determined by shifting those $\Delta H^{SD}(t)/\Delta H^{*SD}(t)$ considering the offset ($\tau$) that is equal to: 1.9 mm, 1.9 mm, 2.6 mm, and 2.0 mm for the mascons #400, #401, #470, and #471, respectively (see Figure 12).

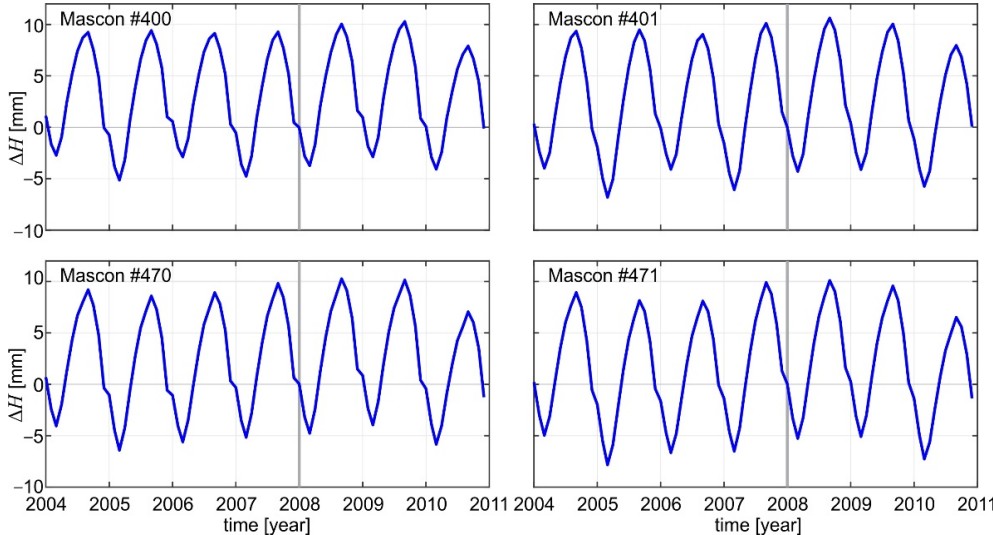

**Figure 12.** Models of temporal variations of orthometric/normal heights ($\Delta H^M/\Delta H^{*M}$) referred to the reference epoch $t_0$, i.e., 01.01.2008.

4.2.3. The Prediction of $\Delta H^M/\Delta H^{*M}$

The GRACE/GRACE–FO level 2 products, i.e., GGMs or mascon solutions, are generally available with a latency of a couple of months from taking measurements by the GRACE/GRACE–FO satellites. Thus, $\Delta H^M/\Delta H^{*M}$ obtained from these products for the present time should be predicted. With the use of $\Delta H^c/\Delta H^{*c}$ data from 6 years preceding the prediction period, $S/S^*$ and $LT/LT^*$ components of $\Delta H/\Delta H^*$ were determined with the SD method. Several mathematical models, e.g., exponential models, Fourier series, Gaussian models, polynomial models, and sum of sines models, were investigated for their fit to the $LT/LT^*$ component of $\Delta H/\Delta H^*$ using Matlab [55]. The 7th degree polynomial model was chosen as the best fitted one. The predicted values of $\Delta H^M/\Delta H^{*M}$ were determined by combing (1) $S/S^*$ component of $\Delta H/\Delta H^*$, (2) values of $\Delta H/\Delta H^*$ resulted from the 7th degree polynomial model fitted to $LT/LT^*$ component of $\Delta H/\Delta H^*$, and (3) the offset ($\tau$).

Figure 13 depicts the example of $\Delta H^M/\Delta H^{*M}$ together with their predicted values $\Delta H^P/\Delta H^{*P}$ as well as $S/S^*$ and $LT/LT^*$ components of $\Delta H/\Delta H^*$. It also illustrates the $LT/LT^*$ component of $\Delta H/\Delta H^*$ obtained using the 7th degree polynomial model. The $\Delta H^P/\Delta H^{*P}$ variations presented in Figure 13 were computed for the period of January 2010–June 2010 using $\Delta H^c/\Delta H^{*c}$ from January 2004 to December 2009. The $\Delta H^c/\Delta H^{*c}$ shifted by $\tau$ and $\Delta H^P/\Delta H^{*P}$, as well as the predicted values of $S/S^*$ and $LT/LT^*$ for this prediction period, were also shown in Figure 13. To obtain $\Delta H^P/\Delta H^{*P}$ in different periods of the year, this procedure was repeated seven times, shifting the beginning of the time series by 1 month. Each time, $\Delta H^P/\Delta H^{*P}$ were predicted for six epochs coinciding with the respective 6 months (Figure 14). The differences

$$\delta P_1 = \Delta H^P(t)/\Delta H^{*P}(t) - \Delta H^{SD}(t)/\Delta H^{*SD}(t) \tag{17}$$

as well as the differences

$$\delta P_2 = \Delta H^P(t)/\Delta H^{*P}(t) - [\Delta H^c(t)/\Delta H^{*c}(t) + \tau] \tag{18}$$

were obtained. The statistic of the differences $\delta P_1$ and $\delta P_2$ for the predicted six months are given in Table 1.

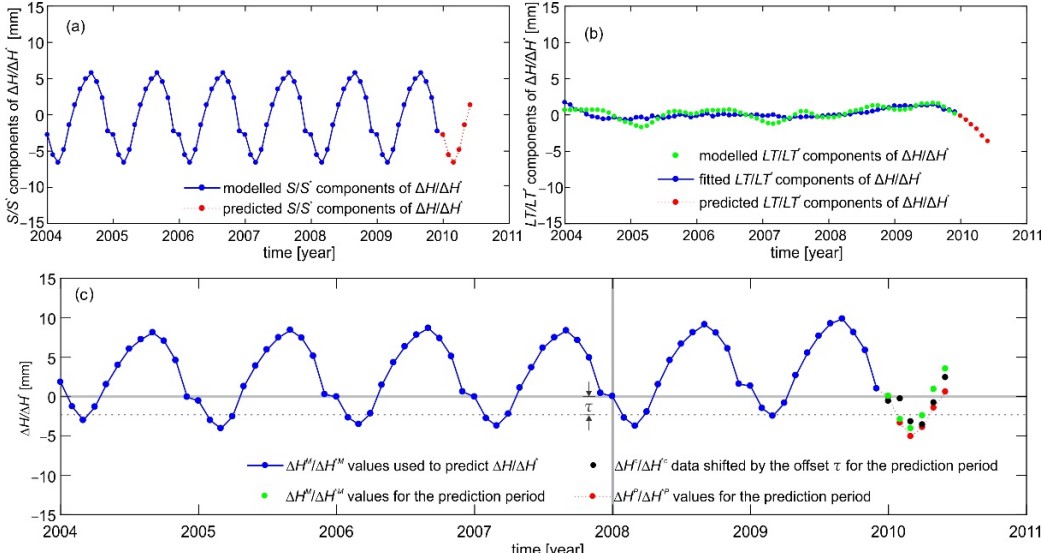

**Figure 13.** Example of (**a**) seasonal components ($S/S^*$) of $\Delta H/\Delta H^*$ and their predicted values, (**b**) long-term components ($LT/LT^*$) of $\Delta H/\Delta H^*$ and their fitted values resulted from the 7th degree polynomial model, and (**c**) $\Delta H^M/\Delta H^{*M}$ model used to predict $\Delta H/\Delta H^*$ together with $\Delta H^c/\Delta H^{*c}$ data shifted by the offset $\tau$, $\Delta H^M/\Delta H^{*M}$ values and $\Delta H^P/\Delta H^{*P}$ values for the prediction period.

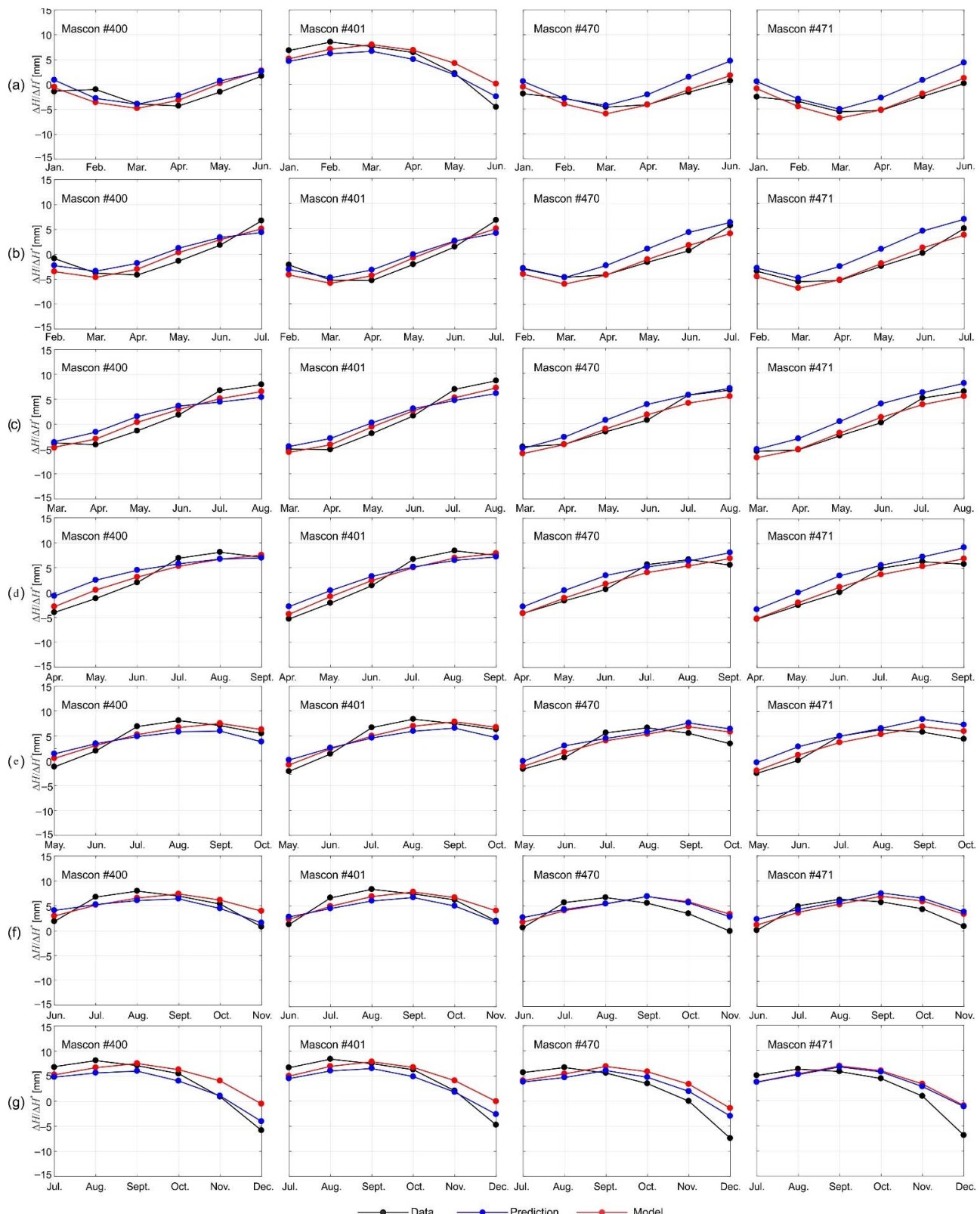

**Figure 14.** The $\Delta H^c / \Delta H^{*c}$ data shifted by the offset $\tau$, $\Delta H^M / \Delta H^{*M}$ obtained from a model of $\Delta H / \Delta H^*$ developed using the seasonal decomposition method and $\Delta H^P / \Delta H^{*P}$. The investigated prediction period (**a**) from January to June 2010, (**b**) from February to July 2010, (**c**) from March to August 2010, (**d**) from April to September 2010, (**e**) from May to October 2010, (**f**) from June to November 2010, and (**g**) from July to December 2010.

**Table 1.** Statistics of differences $\delta P_1$ between $\Delta H^P/\Delta H^{*P}$ and $\Delta H^M/\Delta H^{*M}$ as well as differences $\delta P_2$ between $\Delta H^P/\Delta H^{*P}$ and $\Delta H^c/\Delta H^{*c}$ data shifted by the offset $\tau$ (mm).

| Mascon | Month | $\delta P_1$ | | | | $\delta P_2$ | | | |
|---|---|---|---|---|---|---|---|---|---|
| | | Min | Max | Mean | Std | Min | Max | Mean | Std |
| #400 | 1st | 0.2 | −1.6 | −0.7 | 0.6 | 1.8 | −2.2 | −0.9 | 1.6 |
| | 2nd | 0.7 | −1.3 | −0.7 | 0.8 | 2.3 | −2.6 | −0.5 | 2.0 |
| | 3rd | 1.1 | −1.4 | −0.4 | 1.1 | 2.5 | −2.9 | −0.4 | 2.4 |
| | 4th | 0.6 | −2.1 | −0.9 | 1.1 | 1.4 | −3.7 | −1.1 | 2.3 |
| | 5th | 2.5 | −0.9 | 0.7 | 1.2 | 2.3 | −2.6 | 0.5 | 2.0 |
| | 6th | 2.4 | −1.1 | 0.7 | 1.2 | 1.9 | −2.2 | 0.3 | 1.5 |
| #401 | 1st | 0.6 | −1.1 | −0.4 | 0.6 | 1.3 | −2.3 | −0.7 | 1.2 |
| | 2nd | 0.9 | −1.1 | −0.5 | 0.8 | 2.6 | −2.1 | −0.4 | 1.8 |
| | 3rd | 1.1 | −1.3 | −0.3 | 1.0 | 2.5 | −2.2 | −0.3 | 2.1 |
| | 4th | 0.7 | −1.5 | −0.4 | 0.9 | 1.9 | −2.5 | −0.5 | 2.0 |
| | 5th | 2.1 | −1.0 | 0.6 | 1.1 | 2.4 | −2.3 | 0.6 | 1.9 |
| | 6th | 2.2 | −0.5 | 1.0 | 1.0 | 2.3 | −1.5 | 0.9 | 1.4 |
| #470 | 1st | −1.1 | −2.9 | −1.9 | 0.7 | 0.1 | −4.0 | −2.0 | 1.6 |
| | 2nd | −1.0 | −2.6 | −1.9 | 0.6 | 0.1 | −3.6 | −1.4 | 1.5 |
| | 3rd | −1.0 | −2.1 | −1.6 | 0.3 | 0.3 | −3.1 | −1.2 | 1.4 |
| | 4th | −1.0 | −1.7 | −1.3 | 0.3 | 0.5 | −2.8 | −1.3 | 1.4 |
| | 5th | −0.4 | −1.3 | −0.8 | 0.3 | 1.1 | −3.0 | −1.2 | 1.7 |
| | 6th | 0.5 | −0.9 | −0.1 | 0.5 | 1.4 | −2.9 | −1.0 | 1.8 |
| #471 | 1st | −1.4 | −3.1 | −2.2 | 0.7 | −0.5 | −4.2 | −2.3 | 1.5 |
| | 2nd | −1.7 | −3.3 | −2.6 | 0.7 | −0.6 | −4.4 | −2.3 | 1.5 |
| | 3rd | −1.7 | −2.7 | −2.3 | 0.4 | −0.4 | −3.8 | −2.0 | 1.2 |
| | 4th | −1.9 | −2.3 | −2.1 | 0.2 | −0.6 | −3.4 | −2.1 | 1.2 |
| | 5th | −1.2 | −1.7 | −1.4 | 0.2 | 0.0 | −2.9 | −1.8 | 1.3 |
| | 6th | −0.5 | −1.1 | −0.7 | 0.2 | 0.7 | −2.9 | −1.3 | 1.5 |

The results given in Table 1 and presented in Figures 13 and 14 reveal uneven differences between the predicted $\Delta H^M/\Delta H^{*M}$ and their corresponding $\Delta H^c/\Delta H^{*c}$ data shifted by the offset $\tau$ for all investigated cases. Differences $\delta P_2$ for the first 3 months range from −3.1 to 1.4 mm, when predicting the values of $\Delta H^M/\Delta H^{*M}$ for 6 months from January 2010 to June 2010 using $\Delta H^c/\Delta H^{*c}$ data from January 2004 to December 2009; they range from −2.9 to 0.0 mm, when predicting the values of $\Delta H^M/\Delta H^{*M}$ for the period from March 2010 to August 2010 using $\Delta H^c/\Delta H^{*c}$ data from March 2004 to February 2010. This may indicate that the fit of the predicted values of $\Delta H^M/\Delta H^{*M}$ to $\Delta H^c/\Delta H^{*c}$ data shifted by the offset $\tau$ strongly depends on the magnitude and the character of $\Delta H/\Delta H^*$ within the predicted period. The statistics given in Table 1 indicate that the prediction accuracy in terms of the standard deviations of $\delta P_1$ and $\delta P_2$ ranges from 0.2 to 1.2 mm, and from 1.2 to 2.3 mm, respectively. It shows that the prediction accuracy of $\Delta H/\Delta H^*$ in the case of $\delta P_1$ is higher compared to the case of $\delta P_2$. This might be ascribed to the fact *that* differences $\delta P_2$ include not only the error of the predicting procedure, such as $\delta P_1$, but also the error resulting from the modelling procedure.

### 4.3. The Determination of $H(t)/H^*(t)$

The inverse distance to a power interpolation method was used to interpolate/extrapolate the $\Delta H^M(t)/\Delta H^{*M}(t)$ referred to the epoch of 1 January 2008 at the sites of the ASG-EUPOS network (see Figure 3). Figure 15, showing these changes, exhibits clear variability of $H/H^*$ at the ASG-EUPOS sites in both time and space domains. This variability can reach 20 mm from March to September. Figure 15 also illustrates that in the space domain, the variability of $H/H^*$ over Poland at the same epoch can reach ~2 mm. The $H(t)/H^*(t)$ for the period of 2004–2011 over the area investigated can be obtained by adding the

interpolated/extrapolated values of $\Delta H^M(t)/\Delta H^{*M}(t)$ referred to the epoch 01.01.2008 to the $H(t_0)/H^*(t_0)$ described in Section 4.1.

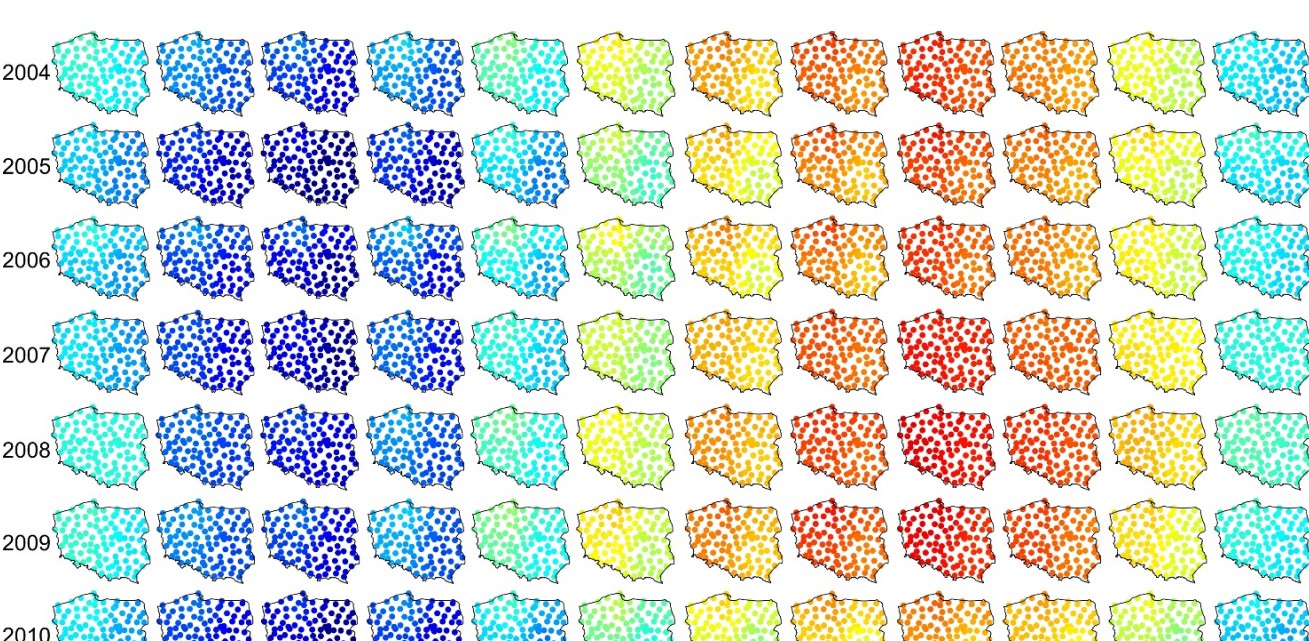

**Figure 15.** Temporal variations of orthometric/normal heights estimated at the ASG–EPOS network sites.

## 5. Conclusions

The paper discusses the determination of orthometric/normal heights $H/H^*$ considering their dynamics of both secular and seasonal character. An approach based on the GRACE satellite mission data are proposed to determine such heights. The results obtained from the implementation of this approach over the area of Poland that was chosen as a case study indicated that:

1.  Temporal variations of orthometric/normal heights $\Delta H/\Delta H^*$ for the period from April 2002 to August 2016, obtained as a combination of temporal variations of geoid/quasigeoid heights $\Delta N/\Delta \zeta$ and vertical deformations of the Earth's surface $\Delta h$ reach up to 23 mm.
2.  The major part of the signal, i.e., ca. 66%, of $\Delta H/\Delta H^*$ results from $\Delta h$, while its remaining part is due to $\Delta N/\Delta \zeta$.
3.  The $\Delta H/\Delta H^*$ are strongly correlated with $\Delta EWT$ (correlation coefficients of $-0.79 \pm 0.03$).
4.  The use of the seasonal decomposition method makes possible modelling $\Delta H/\Delta H^*$ with one millimetre accuracy at the confidence level of 97%; it also makes possible predicting them for the next six months with the accuracy of ca. 1–2 mm.

Overall, the research conducted within the course of the paper emphasizes the need for the $H/H^*$ to be corrected for their dynamics. Such heights will be required for fulfilling the contemporary geodetic scientific purposes and high-precision applications associated with the physical heights. For example, $\Delta H/\Delta H^*$ can particularly be significant to mitigate artifacts and aliasing of repeated levelling measurements. Let us assume that two repeated levelling campaigns were conducted in different epochs over the area of Poland, the first campaign in winter/spring seasons (February–April) and the second campaign in summer/autumn seasons (i.e., August–October). In order to merge and integrate $H/H^*$ determined from these levelling campaigns, $\Delta H/\Delta H^*$ should be considered, otherwise, all levelling measurements associated with the first campaign will be biased by ca. 2 cm

with respect to those from the second campaign. Furthermore, $H/H^*$ are nowadays, more and more frequently determined by combining ellipsoidal heights from GNSS data with geoid/quasigeoid heights obtained from highly accurate gravimetric geoid/quasigeoid model. Taking into the consideration $\Delta N/\Delta \zeta$ and $\Delta h$ in Poland, the dynamics of orthometric/normal heights will essentially be needed for the determination of $H/H^*$ in Poland when using GNSS solutions combined with a precise geoid/quasigeoid model.

The gravity-dedicated satellite missions, e.g., GRACE, provide valuable information for the determination of $H/H^*$ corrected for their dynamics. In particular, they provide unique information concerning long-wavelength components of physical height changes. Since the spatial resolution of $\Delta H/\Delta H^*$ obtained from GRACE satellite mission data is limited to $3° \times 3°$ at the equator, complementary data of high spatial resolution from other sources, e.g., high-resolution hydrological models, would be required to gain more information about short–medium wavelength components, i.e., beyond d/o 60, of $\Delta H/\Delta H^*$.

**Author Contributions:** Conceptualization, M.S., W.G. and J.K.; methodology, M.S. and W.G.; software, W.G.; formal analysis, M.S. and W.G.; data acquisition, M.S. and W.G.; writing—original draft preparation, M.S. and W.G.; writing—review and editing, J.K.; visualization, M.S. and W.G.; supervision, J.K.; project administration, W.G.; funding acquisition, W.G. and M.S. All authors have read and agreed to the published version of the manuscript.

**Funding:** This research was funded by the National Science Center, Poland, grant number: 2021/42/E/ST10/00218.

**Data Availability Statement:** Not applicable.

**Conflicts of Interest:** The authors declare no conflict of interest.

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
