# Peer review of "Contribution of GRACE Satellite Mission to the Determination of Orthometric/Normal Heights Corrected for Their Dynamics—A Case Study of Poland"

_remotesensing, doi:10.3390/rs14174271_

Round 1
Reviewer 1 Report
This paper entitled “Contribution of GRACE satellite mission to the determination of orthometric/normal heights corrected for their dynamics – A case study of Poland ” studied the dynamic change of orthometric/normal heights caused by mass transports, which can be detected by GRACE and GRACE-FO missions. It is an interesting study, and the authors also show a case study in Poland and the importance of mass change in determining heights in this area. Therefore, I recommend a minor revision after carefully checking. Some small places, e.g., 1 mm/km, may be explained scientifically, with accuracy and spatial resolution.
Author Response
The authors would like to thank the reviewer for her/his positive opinion about our article, as well as for her/his valuable comment to improve our manuscript. On the basis of reviewer’s comments, the manuscript has carefully been revised.
In lines 42-43, the sentence “The H/H* of accuracy better than 1 mm/km can be obtained …” has been revised to “The H/H* of the mean square error of the adjusted levelling network less than 1 mm/km can be obtained …”.
Reviewer 2 Report
Dear author, dear editor,
the study about contribution of GRACE satellite mission to the determination of orthometric/normal heights and presentation of approach to determine such heights were widely discussed here. The analysis methods used are well-known but not well-established in determination of orthometric/normal heights over the Poland area. In this respect, the manuscript provides enough new insights to be published.
The manuscript is well organized and reproducible. In my opinion, manuscript may be printed as it is. I rate the scientific level of the manuscript very highly.
In order to improve the manuscript, I have only minor concerns:
-please check abbreviations using in text core, there is some inconsistency, for example ASG-EUPOS is introduce in line 205 without expanding, EUVN in 216 line, GGM in 141 line.
Author Response
The authors would like to thank the reviewer for her/his positive opinion about our article as well as for her/his valuable comment to improve our manuscript. On the basis of reviewer’s comments, the manuscript has carefully been revised. All abbreviations within the context of the manuscript has been revised and defined.
In line 142, “GGM” has been revised to “Global Geopotential Models (GGMs)”.
In lines 208-209, “ASG-EUPOS sites” has been revised to “ASG-EUPOS (Polish Active Geodetic European Position Determination System; http://www.asgeupos.pl/) network sites”.
In line 212, “the ASG-EUPOS (Polish Active Geodetic European Position Determination System) network sites” has been revised to “the ASG-EUPOS network sites”.
In lines 219-220, “EUVN, and EUVN-DA” has been revised to “EUVN (European Vertical Reference Network), and EUVN-DA (European Vertical Reference Network - Densification Action)”.